# ImmunoSep (Personalised Immunotherapy in Sepsis) international double-blind, double-dummy, placebo-controlled randomised clinical trial: study protocol

Antigone Kotsaki,[1] Peter Pickkers [ID],[2] Michael Bauer,[3] Thierry Calandra,[4] Mihaela Lupse,[5] W Joost Wiersinga,[6] Sylvain Meylan [ID],[4] Frank Bloos [ID],[3] Tom van der Poll,[6,7,8] Marleen A Slim [ID],[9] Niels van Mourik,[9] Marcella C A Müller,[9] Lonneke van Vught,[8] Alexander P J Vlaar,[9] Aline de Nooijer,[10] Lieke Bakkerus,[10] Sebastian Weis,[11] Nikolaos Antonakos,[1] Mihai G Netea,[12] Evangelos J Giamarellos-Bourboulis [ID] [1]

MGN and EJG-B contributed equally.

For numbered affiliations see end of article.

**Correspondence to**
Prof. Evangelos J Giamarellos-Bourboulis;
egiamarel@med.uoa.gr

## ABSTRACT

**Introduction** Sepsis is a major cause of death among hospitalised patients. Accumulating evidence suggests that immune response during sepsis cascade lies within a spectrum of dysregulated host responses. On the one side of the spectrum there are patients whose response is characterised by fulminant hyperinflammation or macrophage activation-like syndrome (MALS), and on the other side patients whose immune response is characterised by immunoparalysis. A sizeable group of patients are situated between the two extremes. Recognising immune endotype is very important in order to choose the appropriate immunotherapeutic approach for each patient resulting in the best chance to improve the outcome.

**Methods and analysis** ImmunoSep is a randomised placebo-controlled phase 2 clinical trial with a double-dummy design in which the effect of precision immunotherapy on sepsis phenotypes with MALS and immunoparalysis is studied. Patients are stratified using biomarkers. Specifically, 280 patients will be 1:1 randomly assigned to placebo or active immunotherapy as adjunct to standard-of-care treatment. In the active immunotherapy arm, patients with MALS will receive anakinra (recombinant interleukin-1 receptor antagonist) intravenously, and patients with immunoparalysis will receive subcutaneous recombinant human interferon-gamma. The primary endpoint is the comparative decrease of the mean total Sequential Organ Failure Assessment score by at least 1.4 points by day 9 from randomisation.

**Ethics and dissemination** The protocol is approved by the German Federal Institute for Drugs and Medical Devices; the National Ethics Committee of Greece and by the National Organization for Medicines of Greece; the Central Committee on Research Involving Human Subjects and METC Oost Netherland for the Netherlands; the National Agency for Medicine and Medical Products of Romania; and the Commission Cantonale d'éthique de la recherche sur l'être human of Switzerland. The results will be submitted for publication in peer-reviewed journals.

**Trial registration number** NCT04990232.

## STRENGTHS AND LIMITATIONS OF THIS STUDY

⇒ ImmunoSep is a double-blinded randomised phase 2 clinical trial with double-dummy, placebo-controlled design.
⇒ This is a personalised medicine study to demonstrate the effects of immunotherapy tailored to specific immune endotypes.
⇒ The primary endpoint is the decrease of mean Sequential Organ Failure Assessment score by day 9.
⇒ ImmunoSep also aims for the identification of novel biomarkers through the integrative analysis of -omics of patients' samples.
⇒ This trial is not powered to demonstrate an effect on mortality.

## INTRODUCTION

Sepsis is a life-threatening organ dysfunction that results from the dysregulated host response to an infection.[1] Patients with sepsis-induced dysregulation present a broad spectrum of perturbation ranging from immune hyperactivation to immune suppression. In this respect, approximately 5%–10% of patients present mainly with fulminant hyperinflammation, an entity also known as macrophage activation-like syndrome (MALS),[2] whereas a sizeable minority of other patients have mainly ineffective responses to secondary infections, a condition described as immunoparalysis.[3]



There is accumulating evidence that delivery of targeted immunotherapy for patients who present with these two extremes may improve outcome. Indeed, the post-hoc analysis of a randomised clinical trial (RCT) conducted more than 25 years ago showed that treatment with anakinra, the recombinant antagonist of the interleukin (IL)-1 receptor, provided 30% decrease of 28-day mortality among patients with hepatobiliary dysfunction and disseminated intravascular coagulation who bear phenotypical characteristics compatible with MALS.[4] During the last years, we have suggested that serum ferritin can be an important diagnostic tool for MALS. Studying 5121 patients who were split into one test and into one validation cohort and studying another confirmation cohort from Sweden, it has been found that serum concentrations of ferritin greater than 4420 ng/mL had specificity 97.1% and 98% negative predictive value for the classification of MALS.[2]

On the other hand, the decrease of the expression of the human leucocyte antigen (HLA)-DR expression on the membrane of circulating monocytes is considered one of the hallmarks of immunoparalysis resulting in a dysfunctional immune response, which in turns leads to susceptibility for secondary infections, prolonged hospitalisation and increased mortality.[5] The presence of immunoparalysis in patients with sepsis is associated with at least 50% risk of death in the subsequent 28 days.[6] Evidence from human volunteers subjected to an endotoxin challenge suggests that immunoparalysis can be reversed by recombinant human interferon-gamma (rhIFNγ).[7] In addition, when rhIFNγ was administered in nine patients with septic shock, reversal of immunoparalysis was also achieved.[8]

Based on the existing evidence, experts from five European countries (Germany, Greece, the Netherlands, Romania and Switzerland) designed a double-blind, double-dummy RCT with the aim to deliver personalised immunotherapy as adjunctive treatment to standard of care. The acronym of the trial is ImmunoSep (EudraCT number: 2020-005768-74; ClinicalTrials.gov NCT04990232) and it is funded by the Horizon 2020 programme of the European Union.

## Objectives

ImmunoSep is a randomised placebo-controlled phase 2 clinical trial with a double-dummy design in which the effect of personalised immunotherapy in patients with sepsis and either MALS or immunoparalysis is studied. The primary hypothesis is that the efficacy of immunotherapy in sepsis depends on the specific immune endotype of each patient, and patient stratification for administration of adjunctive immunotherapy aiming to reverse MALS and sepsis-induced immunoparalysis improves chances for a better outcome when compared with a one-size-fits-all immunotherapy approach. It is anticipated that organ dysfunctions as expressed by the mean SOFA (Sequential Organ Failure Assessment) score will be improved by day 9 after randomisation.

## METHODS AND ANALYSIS
### Study design and setting

ImmunoSep is a prospective randomised placebo-controlled phase 2 clinical trial in a total number of 24 academic and non-academic study sites in Greece, Germany, Switzerland, the Netherlands and Romania aiming to assess whether personalised adjunctive immunotherapy directed against a state of either fulminant hyperinflammation or immunoparalysis is able to improve sepsis outcomes. Patients will be selected by a panel of biomarkers and laboratory findings and will be allocated to placebo or immunotherapy treatment according to their needs by 1:1 ratio. The study enrolment will be competitive between the participating study sites targeting 280 participants.

### Study population

Inclusion and exclusion criteria determining the eligibility of study participants are reported in box 1.

### Study procedures

Patients eligible for the study are patients either admitted to hospital from the emergency department or patients already hospitalised in the general clinical wards or in intensive care units. Once a patient is presenting with at least two of the signs of the systemic inflammatory response syndrome or at least one point of the quick SOFA score, then he/she or a legal representative in case the patient cannot consent, is asked for written informed consent. Trial procedures and flow are summarised in figure 1. When the patient does not meet any exclusion criteria, he/she is screened for the presence of lower respiratory tract infection (community-acquired pneumonia, hospital-acquired pneumonia or ventilator-associated pneumonia) or primary bacteraemia and for the Sepsis-3 definition.[1] If the patient meets all these inclusion criteria, the patient is then screened for MALS and immunoparalysis. For this, whole blood is drawn for the measurement of ferritin by an enzyme immunosorbent assay, and for the expression of HLA-DR molecules on CD14+ CD45+ monocytes using the Quantibrite assay by flow cytometry (BD Biosciences, New Jersey, USA). Screened patients will be classified into three groups of immune-endotypes: (a) MALS when serum ferritin is more than 4420 ng/mL irrespective of the number of HLA-DR molecules on circulating CD14+ CD45+ monocytes; (b) immunoparalysis when serum ferritin is 4420 ng/mL or lower and the number of HLA-DR molecules on circulating CD14+ CD45+ monocyte is less than 5000; and (c) unclassified when serum ferritin is 4420 ng/mL or lower and the number of HLA-DR molecules on circulating CD14+ CD45+ monocytes is 5000 or more. If a patient meets the criteria for both MALS and immunoparalysis, MALS is considered as the dominant diagnosis due to the higher mortality in this condition. Patients of immunogroups (a) and (b) may be enrolled in the trial provided they meet the time difference of less than 72 hours from onset of sepsis.

## Box 1 Inclusion and exclusion criteria of the ImmunoSep trial

Inclusion criteria (patients should meet ALL of them)
⇒ Age equal to or above 18 years.
⇒ Both genders.
⇒ In case of women, unwillingness to become pregnant during the study period; birth control measures apply.
⇒ Written informed consent provided by the patient or by one first-degree relative/spouse in case of patients unable to consent.
⇒ Community-acquired pneumonia or hospital-acquired pneumonia or ventilator-associated pneumonia or primary bacteraemia (blood-stream infection (BSI)).
⇒ Sepsis defined by the Sepsis-3 definitions.
⇒ Patients with signs of macrophage activation-like syndrome or sepsis-associated immunoparalysis.
⇒ Time from classification into sepsis by the Sepsis-3 definitions and start of blind intervention less than 72 hours.

Exclusion criteria (patients meeting ANY of the following criteria CANNOT be enrolled)
⇒ Age below 18 years.
⇒ Refusal of written informed consent.
⇒ Acute pyelonephritis or intra-abdominal infection, meningitis or skin infection.
⇒ Any stage IV malignancy.
⇒ Neutropenia defined as an absolute neutrophil count lower than 1500/mm$^3$.
⇒ Any 'do not resuscitate' decision in the hospital.
⇒ In the case of BSI, patients with blood cultures growing coagulase-negative staphylococci or skin commensals or catheter-related infections cannot be enrolled.
⇒ Active tuberculosis (TB) as defined by the co-administration of drugs for the treatment of TB.
⇒ Infection by the HIV.
⇒ Any primary immunodeficiency.
⇒ Oral or intravenous intake of corticosteroids at a daily dose equal to 0.4 mg/kg prednisone or greater for more than the last 15 days.
⇒ Any anti-cytokine biological treatment the last 1 month.
⇒ Medical history of systemic lupus erythematosus.
⇒ Medical history of multiple sclerosis or any other demyelinating disorder.
⇒ Pregnancy or lactation. Women of childbearing potential will be screened by a urine pregnancy test before inclusion in the study.

Study screening is facilitated by the generated web platform at the address https://sepsisonline.org. Investigators blinded to the study intervention enter the data on the inclusion and exclusion criteria in the platform and then the screened patient receives a code. Blinded investigators do not have access to the laboratory results of classification into immunogroups. These laboratory results are sent by an email to the unblinded investigators. Once the unblinded investigators receive the email, they enter the results of ferritin and HLA-DR on the platform and they receive immediate notification if the patient is enrolled or not and, in case of enrolment, of the immunogroup (MALS, sepsis-induced immunoparalysis) and if the patient is allocated to placebo or active treatment. Separate secure usernames and passwords are generated for every blinded and unblinded investigator permitting

restricted access to the platform fields according to their role in the study.

### Study intervention/allocation to blind treatment

The unblinded pharmacists receive information on the daily preparation of the study drug by the site https://sepsisonline.org where they have to log-in using a secure username and password. Once a patient is considered eligible for enrolment, he will be allocated blindly 1:1 to one of the two groups of treatment. The randomisation is stratified by study site and it is done using one computer-generated sequencing. The preparation of the study drug will be done in each study site by the unblinded pharmacist investigators. The two groups of treatment are:

► **Placebo**. In addition to standard-of-care treatment decided by the attending physicians, patients will also receive intravenously 20 mL (10 mL for patients when creatinine clearance is lower than 30 mL/min) 0.9% saline (N/S) three times a day (every 8 hours) for 15 days and 0.5 mL subcutaneously 1 mL 0.9% N/S every other day for a total of 15 days.
► **Active immunotherapy**. In addition to standard-of-care treatment decided by the attending physicians, patients with MALS will receive 200 mg anakinra every 8 hours and subcutaneous placebo as specified above, whereas patients with immunoparalysis will receive intravenous placebo as specified above and subcutaneous 100 μg rhIFNγ once every other day for a total of 15 days. Creatinine clearance should be calculated daily by the Cockcroft Gault equation and when it is lower than 30 mL/min, anakinra will be given at half dose.

### Blinding protocol

The preparation of the study drug will be done in each study site by the unblinded pharmacist investigators. Syringes with active drug or placebo will be covered to conceal the identity of the test article. The unblinded pharmacist will provide the covered syringes to the blinded nurse or blinded investigator who will administer the infusion. All other subinvestigators and the patients are blinded to the assigned intervention.

### Study procedures

An overview of all study procedures is provided in table 1. Briefly, visits 1–15, visit 21, visit 28 and visit 90 include recording of co-administered drugs; SOFA score; vital signs; absolute blood cell count and differentiation, haemoglobin and absolute platelet count (if available); biochemistry (if available); coagulation (if available); blood gasses (if available), microbiology and antibiogram (if available). There would also be recording of the clinical state of the infection making the patient eligible for the study. Of utmost importance is recording and subsequently properly reporting any adverse event/serious adverse event occurring during trial participation. At visit days 1, 2, 4, 7, 15, 21, 28 and 90, blood sampling for trial-related purposes is drawn (transcriptomic, flow cytometry,

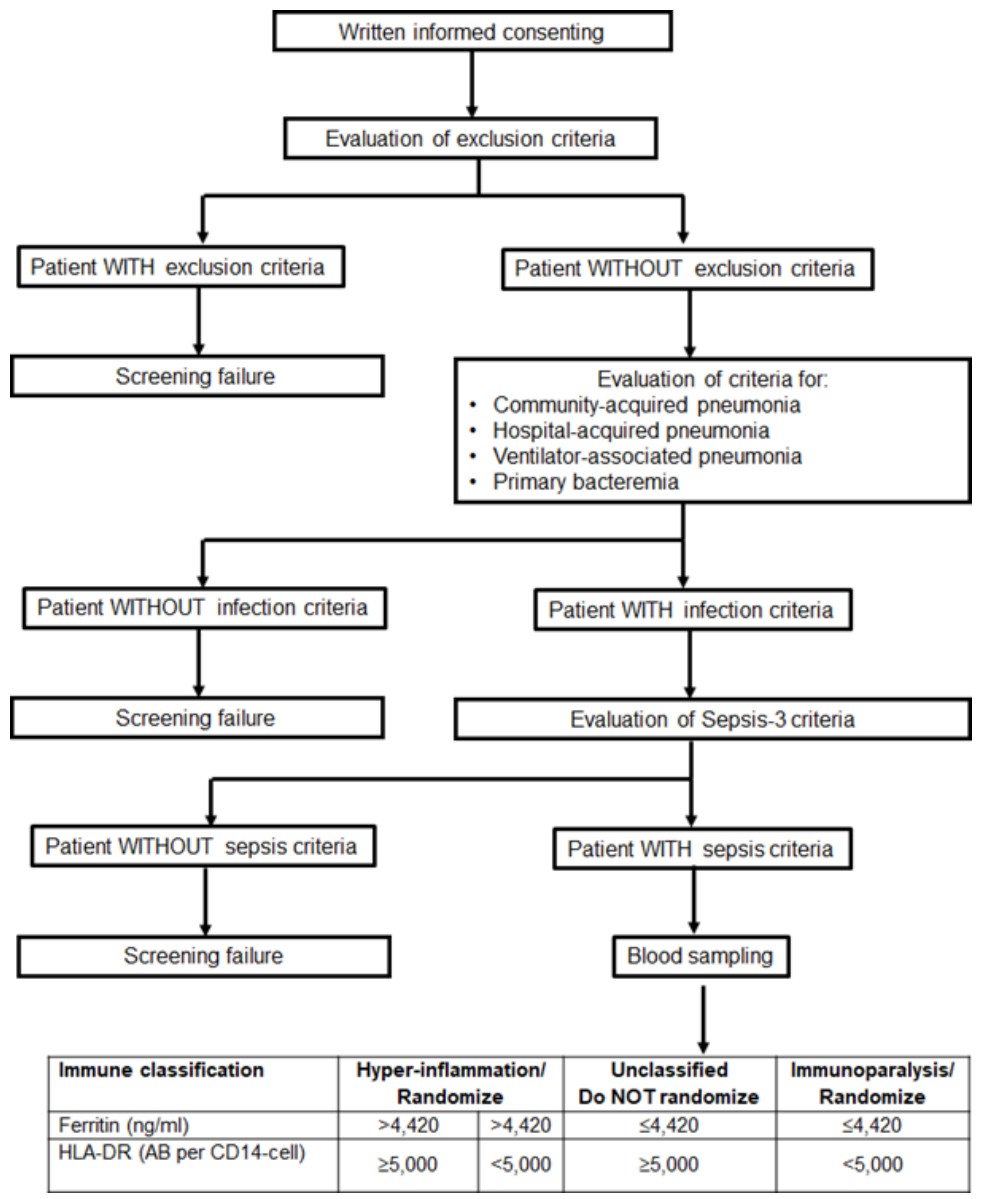

**Figure 1** Screening process for patient eligibility for enrolment in the ImmunoSep Study. HLA, human leucocyte antigen.

cytokine production, ATAC-epigenome sequencing, proteomic analysis) and microbiome samples from nares, oral cavity and rectum. All collected samples are pseudo-anonymised using a 9-digit code. Separate coding is done per study site.

### Study endpoints

The primary efficacy endpoint is the difference in the mean total SOFA score until day 9 after randomisation between the two groups of treatment. The time frame of 9 days for the assessment of the primary endpoint is based on previous experience coming from the VISEP[9] and MAXSEP[10] RCTs where this endpoint was used to assess an impact of treatment on sepsis-induced organ dysfunction at day 9.

The secondary study endpoint is the comparison of the two groups of treatment on: (a) 28-day all-cause mortality; (b) 90-day all-cause mortality; (c) the mean total SOFA score on day 15 from randomisation; (d) the impact of personalised immunotherapy on the reversal of MALS or immunoparalysis on day 15 from randomisation, defined for patients with MALS as at least 15% decrease of the baseline serum ferritin, and for patients with immuno-paralysis as restoration of HLA-DR expression on CD45/CD14 monocytes above 8000/cell. The assessment of the mean total SOFA score on day 15 from randomisation is a read-out of treatment efficacy at the end of treatment. It is considered that a later time point of assessment of the mean total SOFA score is not needed since mortality is the secondary endpoint already assessed at a later time point.

Exploratory study endpoints are: (a) the impact of personalised immunotherapy on the resolution of infec-tion leading to study enrolment on day 15 after randomis-ation; and (b) the development of genomic, epigenomic,

**Table 1** Study visits

| Day | Study visits | | | | | | | | | | | | | | | | | | |
|---|---|---|---|---|---|---|---|---|---|---|---|---|---|---|---|---|---|---|---|
| | 0 | 1 | 2 | 3 | 4 | 5 | 6 | 7 | 8 | 9 | 10 | 11 | 12 | 13 | 14 | 15 | 21 | 28 | 90 |
| Obtain ICF | X | | | | | | | | | | | | | | | | | | |
| Study drug | | X | X | X | X | X | X | X | X | X | X | X | X | X | X | X | | | |
| SOFA score | | X | X | X | X | X | X | X | X | X | X | X | X | X | X | X | X | | |
| Medical history | | X | | | | | | | | | | | | | | | | | |
| Clinical state of infection | | | | | | | | X | X | X | X | X | X | X | X | X | X | X | | |
| Survival | | X | X | X | X | X | X | X | X | X | X | X | X | X | X | X | X | X | X |
| Vital signs | | X | X | X | X | X | X | X | X | X | X | X | X | X | X | X | | | |
| Lab tests | | X | X | X | X | X | X | X | X | X | X | X | X | X | X | X | | | |
| Microbiology | | X | X | X | X | X | X | X | X | X | X | X | X | X | X | X | | | |
| Blood collection | | X | X | | X | | | X | | | | | | | | X | X | X | X |
| Microbiome samples | | X | X | | X | | | X | | | | | | | | X | X | X | X |
| Co-administered medication | | X | X | X | X | X | X | X | X | X | X | X | X | X | X | X | | | |
| Adverse events | | X | X | X | X | X | X | X | X | X | X | X | X | X | X | X | X | X | X |

ICF, informed consent form; SOFA, Sequential Organ Failure Assessment.

proteomic, metabolomic and microbiomic surrogate biomarkers for the primary and secondary endpoints. This will come from the exploitation of the genomic and proteomic material that will be analysed by the partners of the ImmunoSep Project.

## Sample size

The study is powered for the primary endpoint, that is, decrease of mean SOFA score by at least 1.4 points on day 9. In order to calculate the power of the study, the following hypotheses are made: according to data from the previous RCTs in sepsis VISEP[9] and MAXSEP[10] on a total of 1137 patients, there is a significant association between the mean total SOFA score at day 9 and 28-day mortality. A reduction of the primary endpoint by 1.4 points is expected to be associated with a reduction of 28-day mortality. Based on the preliminary results of the PROVIDE Study (ClinicalTrials.gov NCT0333225), 40% of the enrolled patients in each arm will be recruited with fulminant hyperinflammation and another 60% for sepsis-associated immunoparalysis. The study is powered for 90% at the 5% level of significance and the anticipated mean difference in the SD between the two groups will be 3.2. In order to detect this difference of 1.4 points in the mean SOFA score, 117 patients will be needed per trial arm. Considering a drop-out rate of about 15%, a total of 280 patients need to be randomised.

## Statistical analysis

The endpoints of the change of the mean SOFA score will be compared between the two groups of treatment using the Welch's t-test for mean differences. The endpoints of time to an event will be analysed using Cox regression analysis. Analysis will be done in the intention-to-treat population with sensitivity analysis for the per-protocol population. Missing values will be imputed by last observation carried forward.

## Ethics and dissemination

This clinical study falls under Directive 2001/20/EC (Clinical Trials Directive). The protocol was submitted and approved by the National Ethics Committee of Greece (approval 2/21); by the National Organization for Medicines of Greece (approval IS008/21); by the Central Committee on Research Involving Human Subjects for the Netherlands (approval NL76706.091.21); by the Commission Cantonale d'éthique de la recherche sur l'être human of Switzerland (approval 2022-00606); by the German Federal Institute for Drugs and Medical Devices (approval 2022-05-25) and by the Ethics Committee of the Jena University Hospital (approval 2022-2540-AMG-ff); and by the National Agency for Medicine and Medical Products of Romania (approval 129E/29-09-2022). The patients will be included after having provided written informed consent to the investigator. If the patient is not able to understand the information given, he/she can be included if the same procedure is completed by one first-degree relative/spouse/legal representative. After the patient's recovery, he/she will be asked if he/she agrees to continue the trial. Her/his consent will again be necessary for the continuation of the study. No study-related procedure will be performed prior to obtaining written informed consent. For the Netherlands and Germany, a separate deferred consent process for patients unable to consent is followed according to applicable legislation, as described in detail in protocol supplements. The trial shall be governed by the international standards for Good Clinical Practice (GCP) developed by the International Council for Harmonization of Technical Requirements

for Pharmaceuticals for Human Use (ICH), the Directive 2001/20/EC for clinical trials and General Data Protection Regulation (GDPR) 679/2016 (EC). One insurance contract is already active to cover financially any harm which may be caused to an individual as a result of participation in the study.

The authors encourage the timely publication of scientific results in peer-reviewed journals to maximise outreach to the scientific community. All publications will be available in open access. It is anticipated that at least four major publications will be generated. The first publication will cover the results of the clinical trial and the other three publications the results of the analysis of collected biomaterial. The main author list will contain the names of all investigators and subinvestigators who contributed most to the generation of the data. Their rank in the main author list will depend on the level of contribution. A separate list containing all the names of all contributing investigators and subinvestigators in all study sites will also be published in each publication. Scientific events/conferences and other networking events will provide a valuable platform for rapid dissemination of results through oral presentations, posters and personal discussions, fostering active dialogue and direct interaction with other members of the scientific community, and paving the way for future scientific collaborations. A final symposium on immunotherapy in infections will be organised to disseminate results and pave the way for a sustainable uptake of results.

### Data collection/data management
Data will be collected on an electronic case report form (CRF) by a trained investigator or research assistant at each centre that can be found at the address https://sepsisonline.org. Data management will be performed by the Hellenic Institute for the Study of Sepsis (HISS) according to ICH-GCP, European Medicines Agency (EMA)/INS/GCP/454280, EMA/226170/2021 and GDPR-applicable regulations. Clinical trial monitoring will be performed by clinical research associates (CRAs) appointed by HISS. CRAs will ensure protocol adherence and GCP compliance, and maintain regular communication with both sites and sponsor during clinical trial conduct. During monitoring visits, source data verification will be carried out by CRAs and all entries in the CRFs will be compared with the original source documents ensuring data integrity. Separate blind and unblind CRAs will be appointed for the actions of the blinded and unblinded investigators, respectively. HISS is also responsible for the pharmacovigilance of the study and for the reporting of any severe and non-severe treatment-emergent adverse events (TEAEs), as well as for the reporting of any serious unexpected severe adverse reactions (SUSARs). All SUSARs are immediately reported to all study sites and to the ethics committees of all involved hospitals and to the committees which approved the study. An annual report of all TEAEs and SUSARs is also provided to all study sites and to the ethics committees of all involved hospitals and

to the committees which approved the study. HISS will also organise audits to the top recruiting study sites by an independent third body.

One Data Safety Monitoring Board (DSMB) is active for the ImmunoSep trial since January 2022. This is composed of Professors Djillali Annane, Antonio Artigas and Adam Linder. The DSMB is planned to monitor the overall safety profile of the study when the follow-up of the first 140 patients will finish. The DSMB will decide on study continuation. Emergency unblinding for safety purposes is allowed after detailed explanation by the principal investigator.

HISS will have access to the final dataset. Access to the dataset is allowed only after contractual agreement.

### Patient and public involvement
Patients or the public were not involved in the design of the ImmunoSep trial.

## DISCUSSION
Risk stratification and delivery of immunotherapy tailored to the needs of every patient are the backbone of the ImmunoSep RCT. ImmunoSep is already running in Germany, Greece, the Netherlands, Romania and Switzerland. As of 1 June 2022, 153 patients were screened and 65 patients were enrolled.

A study of precision immunotherapy tailored to the needs of every patient has two main requirements: (a) the mechanism driving the immune dysfunction of the host is well defined, and (b) the immune state of the enrolled patients is driven by an immune endotype that can be identified by readily available biomarkers. This is the reason why only patients with sepsis who are suffering from well-defined MALS or immunoparalysis are randomised to receive immunotherapy in the ImmunoSep trial, whereas unclassified patients who do not fulfil the immunological criteria are not enrolled.

There are major challenges in running the ImmunoSep trial, which may be summarised as follows: (a) the anticipated screening failure rate and (b) the utility of the primary endpoint. The two biomarkers used, ferritin and the absolute count of HLA-DR molecules on CD45/CD14 monocytes, have been used for immune classification in the previously run trial PROVIDE (ClinicalTrials.gov NCT03332225). Published results in study participants with community-acquired pneumonia validated the ability of the biomarkers to classify patients into MALS, immunoparalysis or unclassified.[11] With the use of these biomarkers, the anticipated screening failure rate based on PROVIDE Study and the initial months of recruitment in ImmunoSep is anticipated to amount between 30% and 60%. Sepsis organ dysfunction is measured through the SOFA score.[1] As such, the introduction of mean SOFA score as an endpoint is reflecting the ability of immunotherapy on restoration of sepsis-induced organ dysfunction.

ImmunoSep is the first study employing patient stratification and precision medicine in immunotherapy for sepsis, and it is anticipated that such an approach has much better chances to improve the outcome of the patients compared with earlier one-size-fits-all clinical trials. Although similar interventions of precision medicine have not yet been performed and registered for sepsis, anakinra has recently been licensed by the EMA for adults with pneumonia by the SARS-CoV-2 coronavirus. Treatment is guided by circulating concentrations of the biomarker suPAR (soluble urokinase plasminogen activator receptor) of 6 ng/mL or more, which is an indicator of the early activation of the IL-1 cascade.[12] Using such precision medicine approach, anakinra treatment was able to significantly decrease 0.36 times the risk for a worse score on the 11-point WHO-Clinical Progression Scale at day 28 (95% CI 0.26 to 0.50, p<0.0001) compared with patients receiving placebo.[13] So, a trial using an elevated suPAR concentration as an enrolment criterion showed improved outcomes, while trials not using an enrichment strategy did not,[14] plausibly reflecting the relevance of phenotyping.

Several recent data generate hope that the administration of anakinra and rhIFNγ may improve outcomes for patients with critical COVID-19. In a recent open-label trial, patients with COVID-19 pneumonia classified with MALS using ferritin more than 4420 ng/mL were treated for 7 days with intravenous anakinra 200 mg every 8 hours; mortality was decreased compared with historical comparators.[15] In five patients with COVID-19 with persistently low HLA-DR expression and incapacity to eliminate the virus, subcutaneous treatment with rhIFNγ led to considerable viral elimination, clinical improvement and discharge from the intensive care unit.[16]

ImmunoSep is a promising approach aiming to change clinical practice for the management of the critically ill patients with sepsis by using patient stratification and precision medicine. Appropriate identification of immune-endotypes with biomarkers and delivery of treatment tailored to patients is likely to represent the future of adjuvant immunotherapy for sepsis and other severe infections.

**Author affiliations**
[1] 4th Department of Internal Medicine, National and Kapodistrian University of Athens, Athens, Greece
[2] Intensive Care Medicine, Radboudumc, Nijmegen, The Netherlands
[3] Klinik für Anästhesiologie und Intensivtherapie, Universitätsklinikum Jena, Jena, Germany
[4] Department of Infectious Diseases, CHUV, Lausanne, Switzerland
[5] Department of Internal Medicine, University of Cluj-Napoca, Cluj, Romania
[6] Department of Internal Medicine, Amsterdam University Medical Centres, Duivendrecht, The Netherlands
[7] The Center of Experimental and Molecular Medicine, Amsterdam University Medical Centres, Duivendrecht, The Netherlands
[8] Department of Infectious Diseases, Amsterdam University Medical Center, Amsterdam, The Netherlands
[9] Intensive Care Unit, Amsterdam UMC Locatie AMC, Amsterdam, The Netherlands
[10] Department of Internal Medicine and Infectious Diseases, Radboud University Medical Center, Nijmegen, The Netherlands
[11] Center for Infectious Disease and Infection Control, Jena University Hospital, Jena, Germany
[12] Radboud University Medical Center, Nijmegen, The Netherlands

**Contributors** AK wrote the current manuscript, edited the protocol, provided feedback for intellectual content for this manuscript and gave final approval for this manuscript to be published. PP, MB, TC, ML, JW, SM, FB, TvdP, MAS, NvM, MCAM, LvV, AV, AdN, LB, SW, NA and MGN edited the protocol, provided feedback for intellectual content for this manuscript and gave final approval for this manuscript to be published. EJG-B wrote the protocol, provided feedback for intellectual content for this manuscript and gave final approval for this manuscript to be published.

**Funding** This work was supported by Horizon 2020 programme grant (847422).

**Competing interests** EJG-B has received honoraria from Abbott, bioMérieux, Brahms, GSK, InflaRx, Sobi and XBiotech; independent educational grants from Abbott, AxisShield, bioMérieux, InflaRx, Johnson & Johnson, MSD, Sobi and XBiotech; and funding from the Horizon2020 Marie-Curie Project European Sepsis Academy (granted to the National and Kapodistrian University of Athens), and the Horizon 2020 European Grants ImmunoSep and RISKinCOVID (granted to the Hellenic Institute for the Study of Sepsis). MGN is a scientific founder and member of scientific advisory board of Trained Therapeutics Discovery and Lemba.

**Patient and public involvement** Patients and/or the public were not involved in the design, or conduct, or reporting, or dissemination plans of this research.

**Patient consent for publication** Not required.

**Provenance and peer review** Not commissioned; externally peer reviewed.

**ORCID iDs**
Peter Pickkers http://orcid.org/0000-0002-1104-4303
Sylvain Meylan http://orcid.org/0000-0001-6319-2423
Frank Bloos http://orcid.org/0000-0002-0767-7941
Marleen A Slim http://orcid.org/0000-0002-6281-040X
Evangelos J Giamarellos-Bourboulis http://orcid.org/0000-0003-4713-3911

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
