## [Reviewer comments · BMJ Open]

ARTICLE DETAILS

TITLE (PROVISIONAL)	The ImmunoSep (Personalized Immunotherapy in Sepsis) international double-blind, double-dummy, placebo-controlled randomized clinical trial: study protocol
AUTHORS	Kotsaki, Antigone; Pickkers, Peter; Bauer, Michael; Calandra, Thierry; Lupse, Mihaela; Wiersinga, Joost; Meylan, Sylvain; Bloos, Frank; van der Poll, Tom; Slim, Marleen; van Mourik, Niels; Müller, Marcella; van Vught, Lonneke; Vlaar, Alexander; de Nooijer, Aline; Bakkerus, Lieke; Weis, Sebastian; Antonakos, Nikolaos; Netea, M; Giamarellos-Bourboulis, Evangelos

VERSION 1 – REVIEW

REVIEWER	Gensheng Zhang Second Affiliated Hospital, Zhejiang University School of Medicine, Critical Care Medicine
REVIEW RETURNED	13-Sep-2022

GENERAL COMMENTS	The MS “The ImmunoSep (Personalized Immunotherapy in Sepsis) international double blind, double-dummy, placebo-controlled randomized clinical trial: study protocol” is interesting and meaningful, some suggestions are list below, (1)Although some explanations were described in the discussion section, the protocol pointed out “The primary endpoint is the comparative decrease of the mean total SOFA (Sequential Organ Failure Assessment) score by at least 1.4 points by day 9 from randomization.”; The question is why the intervention by Personalized Immunotherapy could improve the SOFA score by day 9, while the Personalized Immunotherapy continue to day 15. Whether the benefit on the improvement of SOFA score by day 16 or at some day longer than day 9 would display more significant or not ? That is to say, which point the Personalized Immunotherapy could improve the organ function of sepsis via direct action or indirect action by this kind of Personalized Immunotherapy? (2)In terms of the three immune endotypes for these screened patients, how to define the situation when the patients meet this kind of immune endotype: Immunoparalysis with serum ferritin 4,420 ng/ml or lower, while large than 5,000 HLA-DR molecules on circulating CD45/CD14-monocyte?
--

REVIEWER	Rujipat Samransamruajkit Chulalongkorn UNiversity, Pediatric Critical Care
REVIEW RETURNED	08-Oct-2022

GENERAL COMMENTS	It is a very interesting multicenter study involve Immunotherapy in adult sepsis (ImmunoSep) It is a prospective randomized placebo-controlled phase 2 clinical
---

	trial in a total number of 24 study sites in Greece, Germany, Switzerland, the Netherlands and Romania aiming to assess whether personalized adjunctive immunotherapy directed against a state of either fulminant hyperinflammation or immunoparalysis is able to improve sepsis outcomes. The answer of this study may result in improvement of adult sepsis who has hyperinflammatory reaction.
--	---

VERSION 1 – AUTHOR RESPONSE

Reviewer: 1 (Dr. Gensheng Zhang, Second Affiliated Hospital, Zhejiang University School of Medicine:

- *The MS “The ImmunoSep (Personalized Immunotherapy in Sepsis) international double blind, double-dummy, placebo-controlled randomized clinical trial: study protocol” is interesting and meaningful, some suggestions are list below.)Although some explanations were described in the discussion section, the protocol pointed out “The primary endpoint is the comparative decrease of the mean total SOFA (Sequential Organ Failure Assessment) score by at least 1.4 points by day 9 from randomization.”; The question is why the intervention by Personalized Immunotherapy could improve the SOFA score by day 9, while the Personalized Immunotherapy continue to day 15. Whether the benefit on the improvement of SOFA score by day 16 or at some day longer than day 9 would display more significant or not ? That is to say, which point the Personalized Immunotherapy could improve the organ function of sepsis via direct action or indirect action by this kind of Personalized Immunotherapy?*

Reply: We wish to thank you for your comment and for the opportunity that you provide us to clarify the endpoints. Regarding the primary endpoint, the revised manuscript now reads on pg 13, In4-7: “The time frame of 9 days for the assessment of the primary endpoint is based on previous experience coming from the VISEP⁹ and MAXSEP¹⁰ randomized clinical trials where this endpoint was used to assess an impact of treatment on sepsis-induced organ dysfunction at day 9.” Regarding the secondary endpoint, the revised manuscript now reads on pg13, In14-17: “The assessment of the mean total SOFA score on day 15 from randomization is a read-out of treatment efficacy at the end of treatment. It is considered that a later timepoint of assessment of the mean total SOFA score is not needed since mortality is the secondary endpoint already assessed at a later timepoint.”

- *In terms of the three immune endotypes for these screened patients, how to define the situation when the patients meet this kind of immune endotype: Immunoparalysis with serum ferritin 4,420 ng/ml or lower while large than 5,000 HLA-DR molecules on circulating CD45/CD14-monocyte?*

Reply: We thank you for this comment. The classification system is now better clarified on pg 9, In6-12 of the revised manuscript that now reads: “Screened patients will be classified into three groups of immune-endotypes: (a) MALS when serum ferritin is more than 4,420 ng/ml irrespective the number of HLA-DR molecules on circulating CD14+CD45+monocytes; (b) immunoparalysis when serum ferritin is 4,420 ng/ml or lower and the number of HLA-DR molecules on circulating CD14+CD45+monocyte is less than 5,000; and (c) unclassified when serum ferritin is 4,420 ng/ml or

lower and the number of HLA-DR molecules on circulating CD14+CD45+monocytes is 5,000 or more.”

Reviewer 2 (Dr. Rujipat Samransamruajkit, Chulalongkorn University):

- *It is a very interesting multicenter study involve Immunotherapy in adult sepsis (ImmunoSep) It is a prospective randomized placebo-controlled phase 2 clinical trial in a total number of 24 study sites in Greece, Germany, Switzerland, the Netherlands and Romania aiming to assess whether personalized adjunctive immunotherapy directed against a state of either fulminant hyperinflammation or immunoparalysis is able to improve sepsis outcomes. The answer of this study may result in improvement of adult sepsis who has hyperinflammatory reaction.*

Reply: We thank this reviewer for his favorable comments

VERSION 2 – REVIEW

REVIEWER	Gensheng Zhang Second Affiliated Hospital, Zhejiang University School of Medicine, Critical Care Medicine
REVIEW RETURNED	30-Nov-2022
GENERAL COMMENTS	The authors answered all of my questions provided for the first review.